# Implementing and Delivering Youth Mental Health Services: Approaches Taken by the Australian Primary Health Network ‘Lead Sites’

**DOI:** 10.3390/ijerph191710494

**Published:** 2022-08-23

**Authors:** Sanne Oostermeijer, Michelle Williamson, Angela Nicholas, Anna Machlin, Bridget Bassilios

**Affiliations:** Melbourne School of Population and Global Health, The University of Melbourne, Melbourne, VIC 3010, Australia

**Keywords:** mental health services, health care reform, adolescent, mental disorders, service uptake, service implementation, youth mental health

## Abstract

Aim: This paper aims to report on effective approaches for, and early impacts of, implementing and delivering services for youth with, or at risk of, severe mental illness commissioned by 10 Primary Health Network (PHN) Lead Sites (reform leaders) in Australia. Methods: The following qualitative data sources were analyzed using a thematic approach: focus group consultations with 68 Lead Site staff and 70 external stakeholders from Lead Site regions; and observational data from one Lead Site meeting with a focus on services for youth with, or at risk of, severe mental illness and one national symposium that was attended by Lead Site staff and service providers. Results: The Lead Site staff described common effective strategies for implementing and delivering youth enhanced services as follows: building on existing youth services, establishing effective linkages with other local youth enhanced services, and providing complementary clinical and non-clinical services. Early impacts of youth enhanced services that were described by Lead Site staff and external stakeholders included: improved service quality and access, positive effects on consumers and/or carers (e.g., reduced symptomology), and sector-wide impacts such as improved service integration. Staff members from two Lead Sites also mentioned negative impacts (e.g., uncertainty of continued funding). Suggestions for future improvements by Lead Site staff and external stakeholders included: involving young people in service design and planning, improving service access, addressing clinical workforce shortages, improving data collection and usage, and establishing greater service integration. Conclusions: These findings highlight the necessity for collaborative and localized responses as well as service models that combine clinical and non-clinical care to address the needs of young people with, or at risk of, severe mental illness. Early impacts that were reported by stakeholders indicated that PHN-commissioned youth-enhanced services had positive impacts for consumers, carers, and the wider service sector.

## 1. Introduction

Youth mental health problems are a global public health concern and affect 10–20% of young people worldwide [1]. In Australia, recent estimates indicate almost 40% of young people experienced a mental health disorder in the 12 months prior [2]. It has long been recognized that young people need tailored responses and approaches to healthcare, including mental health care services [3]. Research has shown that young Australians typically use mental health services less frequently compared to the overall population [4,5]. It has been argued this is partly because mental health services do not sufficiently meet their needs [6]. Research suggests that young people mainly benefit from flexible mental health service models (i.e., outreach, after-hour services and various modes of service delivery, e.g., internet-based services) and integrated clinical and non-clinical services [7,8]. Tailoring a more youth-specific service model involving accepting referrals from a wide variety of other services and immediate access to secondary care, has been shown to result in a greater uptake of services among young people (e.g., psychiatry, clinical psychology, and vocational services) [9]. A recent scoping review identified key requirements for youth-specific mental health services (youth ‘service hubs’ mostly across high-income Western countries) including: an emphasis on rapid access to care and early intervention, youth and family engagement, youth-friendly and non-stigmatizing settings and services, evidence-informed approaches, and partnerships and collaboration with a diverse set of industry stakeholders [10]. Young people themselves have indicated that, in general, the relationship with service providers and the design and location of services are particularly important when aiming to provide ‘youth-friendly’ services [11]. However, young people of different ages, cultural backgrounds, and gender may have different interpretations and definitions of what is youth-friendly for them [11]. Therefore, successfully engaging different groups of young people in youth-friendly mental health services is likely to involve a range of different approaches.

The *headspace* program in Australia is a well-known example of services that are tailored to the needs of young people. The program typically provides services for young people that are aged 12 to 25 years with mild to moderate mental illness and typically have a youth-friendly and early intervention focus, comprising of mental health, alcohol and other drugs, physical health, and vocational services. Previous research across the first 30 *headspace* centers in Australia has shown a wide variety of activities across the centers in order to achieve community engagement, local partnerships with other services, service integration, and youth participation [12]. This highlights that youth services likely benefit from a place-based approach that responds to local community needs, resources, and capacities. An independent review of the headspace program in 2015 found small positive improvement in the outcomes of young people who sought services from the *headspace* program [13]. To further improve service uptake by young people and provide ‘youth-friendly’ services, a better understanding of effective implementation and delivery approaches for youth services is needed.

### Mental Healthcare Reform in Australia

Primary mental healthcare services play an important role in the prevention and early intervention efforts to reduce the prevalence and impact of mental health problems amongst young people. In response to a major review in 2014 [14], the Australian Government committed to reform the primary healthcare system to improve the efficiency, effectiveness, and coordination of primary care services. Subsequently, the Government initiated the establishment of 31 Primary Health Networks (PHNs) nationwide to lead these reforms within their region, based on local primary healthcare needs [15]. From July 2016, PHNs received funding from the Australian Government Department of Health to plan and commission services in six national mental health priority areas (‘PHN-led reform’), including child and youth services [16].

A total of 10 PHNs (i.e., Lead Sites) were selected to act as mental health reform leaders and provide enhanced services in several nominated key focus areas, including services for youth with, or at risk of, severe mental illness (i.e., youth enhanced services), the focus of this paper. PHNs were required to commission enhanced services to meet the needs of youth with, or at risk of, severe mental illness but have flexibility to determine how these services are delivered in their region based on local needs. The Department of Health commissioned researchers from the University of Melbourne to evaluate the implementation of these mental health reforms [17]. This included the evaluation of approaches to the implementation and delivery of youth enhanced services specifically.

To inform a better understanding of effective implementation of mental health services for young people, this paper reports on effective approaches for, and early impacts of, implementing and delivering of youth enhanced services that were commissioned by 10 PHN Lead Sites (reform leaders) in Australia.

## 2. Methods

Approval was obtained from the Human Ethics Sub-Committee at the University of Melbourne (1749426). The preliminary findings discussing the main challenges of, and facilitating factors to, implementing youth enhanced services during the first year and a half (from 1 July 2016 to 31 December 2017) of PHN activities have been described elsewhere [18]. The current study involves data collection approximately one year later and describes the effective approaches for, and early impacts of, the implementation and delivery of youth enhanced services in Australia. Several qualitative data sources were used, which were collected from September 2018 to March 2019. These included stakeholder consultations with Lead Site staff and external stakeholders in Lead Site regions, and observational data from one Lead Site meeting and a national Lead Site symposium. These are discussed in more detail below.

### 2.1. Stakeholder Consultations

Semi-structured consultations were held with staff from Lead Sites and external stakeholders (e.g., individual providers or service provider representatives) in Lead Site regions that were involved in the planning, implementation and/or delivery of PHN commissioned services. All the participants were provided a plain language statement, a consent form, and asked a set of demographic questions. Lead Site representatives received the consultation questions in advance of the consultations and were asked to invite staff that were best able to answer the questions to participate. The Lead Site representatives were also asked to review the list of 10 to 20 external regional stakeholders who participated in an earlier round of consultations [18], to revise their contact details, and provide details for additional stakeholders from their region. There were no exclusion criteria for participation in the consultations.

The consultations were conducted via separate focus groups for Lead Site staff and external stakeholders. The participants were also given the option of providing written responses if this was their preferred form of participation, or if they were not able to attend the focus group. Pre-determined questions were asked about the implementation and early impacts of youth enhanced services, and suggestions for future improvements. The questions varied for Lead Site staff and external stakeholders (see Appendix A). Lead Site staff were asked to describe any new youth enhanced services that were commissioned in the past 12 months, or any changes to pre-existing services; and effective approaches for the implementation and delivery of youth enhanced services. External stakeholders were asked the effects of new youth enhanced services and how they might be improved in the future.

A total of 10 Lead Site focus groups were held between September and December 2018, which were 2 h and 15 min to 3 h in duration. One focus group was partly conducted via Zoom, an online platform for video conferencing. All other focus groups were conducted face-to-face. There were two members from one Lead Site that participated in separate individual telephone interviews, since they were unable to attend the focus group in person. These two telephone interviews were approximately 30 min each in duration and took place in December 2018. All the recordings from the focus groups were professionally transcribed verbatim. The individual telephone interview recordings were summarized by one of the researchers who took notes during the interviews.

The external stakeholder consultations took place between February and March 2019. A total of eight focus groups were conducted via Zoom and were 1 h to 2 h and 15 min in duration. Notes were taken during the verbal consultations and recordings were used to add to these notes as needed.

All the consultations were video- or audio-recorded. Transcripts, notes, and written responses to the consultations were used for analyses.

### 2.2. Observational Data

There were two researchers with experience in themed notetaking that attended a Lead Site meeting that was held for Lead Site representatives in September 2018 and a two-day Lead Site symposium held for Lead Site representatives and service providers in March 2019. Both events specifically focused on youth enhanced services. The researchers took notes about the key themes around youth enhanced services that transpired during these events.

### 2.3. Data Analyses

Data from the consultations with Lead Site staff and external stakeholders and the observational data, were analyzed using NVivo 12. Each transcript was coded with an initial template that was based on the pre-determined questions focusing on youth enhanced services (Appendix A). The two researchers deductively identified data that were related to the implementation and delivery of youth enhanced services using a thematic analysis approach [19]. Each researcher coded one transcript independently, iteratively creating a coding template. Once the final coding template was constructed, the transcripts were re-examined and the researchers inductively identified narrower sub-themes. A framework analysis approach was used to order the themes in an overarching framework to make sense of the emerging themes [20]. The two researchers compared and revised the coding template until a consensus was reached. The final coding template was applied across all the relevant stakeholder data.

## 3. Results

### 3.1. Sample and Demographic Information

A total of 68 Lead Sites staff participated in the consultations, ranging from 2 to 12 staff members per site. Most of the participants were female (75%) and the mean age was 43.2 years (S.D. 9.3). None of the participants were Aboriginal and/or Torres Strait Islander peoples. Most staff members participated in a focus group (97%) and few via a telephone interview (3%). The Lead Site staff generally comprised of a senior PHN mental health manager and a person representing each portfolio within the mental health stream; for example, managers or program officers for youth mental health, suicide prevention, service intake, or alcohol and other drugs. Some participants had broader responsibility for mental health services in general. Other participants included those that were responsible for evaluation and research, data and planning, policy, and system re-development.

In total, 70 external stakeholders participated in consultations (five to eight per Lead Site). Most external stakeholders were involved with one of the Lead Sites (74.3%), some with two or three Lead Sites (14.3%), one with seven Lead Sites (1.4%), and one with all ten Lead Sites (1.4%). This information was missing for six participants (8.6%). Most of the participants were female (54%) and the mean age was 49.9 years (S.D. 9.4). One participant identified as Aboriginal or Torres Strait Islander. Most external stakeholders participated via a focus group (90%) and some provided written feedback (16%), for four participants, written feedback was in addition to partial focus group participation. Most stakeholders (74.3%) were managers, CEOs, or employees from external service provider agencies (including five Local Health Networks [LHNs]). The remainder included representatives from professional and/or peak bodies, representatives from the local health department and mental health commission, and independent consultants and researchers.

The Lead Site meeting was attended by Lead Site staff and service providers that were commissioned by three Lead Sites with a specific focus on youth enhanced services. The national symposium was attended by 30 of the 31 PHNs and many service providers that were involved in child and youth services.

### 3.2. Youth Enhanced Services Commissioned by Lead Sites

Staff from seven Lead Sites described youth enhanced services and/or activities that were new or had undergone changes in the previous 12 months. These included: developing a consortium-based model to run services across four areas in one Lead Site region, co-locating services with *headspace*, commissioning assertive outreach and some alcohol and other drug services, and broadening age group eligibility (from 12–18 years to 12–25 years) to access services. Furthermore, staff described the implementation of services specifically targeting young people at risk of being suspended from school due to substance misuse, a new service model with a multidisciplinary team to conduct outreach work, and a hospital-based functional recovery service for youth with early psychosis (i.e., a separate program to the Early Psychosis Youth Services program). Finally, specifically targeting youth with complex mental health needs was also described. Notably, during the Lead Site meeting three of the five youth enhanced service providers described specifically focusing on ‘disengaged’ young people and providing outreach.

#### 3.2.1. Effective Approaches for Implementing and Delivering Youth Enhanced Services

##### Building on Existing Youth Services

Staff from four Lead Sites reported that building upon existing local youth services was an effective implementation strategy because the services were already embedded in their communities. As a result, the Lead Sites had existing relationships with important stakeholders that led to ease of referrals and a better ability to engage with disengaged young people within their communities. One staff member illustrated this by commenting:


*‘The target group [disengaged young people] was already turning up to the agency we commissioned. They are really well known by the community, young people trusted this service, good track record of the service. If we would have started up a new service, it probably would have been a lot slower.’*


##### Linkages with Other Youth Enhanced Services

Staff from four Lead Sites described facilitating service integration by establishing effective linkages with other youth enhanced services within their community. These linkages were created using various approaches including co-locating youth enhanced services within youth services and a general practice; establishing a consortium of service providers to deliver youth enhanced services; and fostering good relationships with local *headspace* centres, the Local Health District (LHD) and the tertiary-level child and adolescent mental health services (CAMHS). One staff member commented:


*‘What worked well was that we had set up early on the view that we’re looking at our [youth service] as youth mental health more holistically across the region, and not looking at [it] as a standalone thing that didn’t integrate with the rest of the service system.’*


##### Complementary Non-Clinical Support

Staff from four Lead Sites gave examples of clinical care being successfully complemented by non-clinical care including vocational, educational, and parental support. Examples of such complementary services included homelessness services, parenting education, and service navigation support to facilitate engagement of disengaged young people and linkages to vocational support. During the Lead Site meeting, it was apparent that all service models that were commissioned by these Lead Sites had a holistic approach and combined clinical and non-clinical treatment components (i.e., wrap-around services). Aligned with this, during the external stakeholder consultations, nine participants from six Lead Site regions were very positive about the ability to provide wrap-around services for young people using collaborative and multi-agency service models. These stakeholders described complementary services being part of the wrap-around model for young people; which included vocational, alcohol and other drugs, and family support services. One stakeholder commented:


*‘Offering a full suite of services at our centres is critical when considering the holistic needs of young people.’*


##### Site-Specific Approaches

A single Lead Site described other effective approaches or specific factors that facilitated the implementation of commissioned youth enhanced services in their local area. This included a variety of approaches relating to the early stages of a service implementation including: involving the LHN early in consultation and in procurement to promote ownership over the new service, co-designing youth enhanced services with a child and youth advisory group, having information sessions with tenderers to clarify the selection criteria, allowing the provider sufficient time for refining the service model before service delivery commencement, developing a strong clinical governance framework with ongoing monitoring by the PHN, and identifying a service that really fills a local service gap. Approaches relating to partnerships or collaborations included: collaboration with local schools to reach young people and working closely with providers to understand and help overcome challenges in service establishment. Workforce-related approaches included getting expert assistance (from Orygen, an Australian research institute which focuses on the prevention and treatment of mental disorders in young people) with workforce development and focusing on workforce development to ensure clinicians delivering youth enhanced services are well-trained to work with young people. Other service approaches included: implementing assertive outreach within the youth enhanced service, replicating existing services in additional locations, defining and specifying eligibility criteria for youth to access the services, using flexible service models that have no limitations on the number of sessions that are provided, and involving multi-disciplinary teams including a family therapist and peer worker.

#### 3.2.2. Early Impacts of Youth Enhanced Services

##### Improved Services and Service Access

Both Lead Site staff and external stakeholders described an improvement in the services that were available and/or better service access for young consumers. Lead Site staff from three sites noted youth enhanced services had resulted in the availability of different types of services. Another Lead Site staff member noted youth enhanced services had filled a local service gap.

Several external stakeholders (five stakeholders from four Lead Site regions) described positive impacts that were related to improved service quality and access including: the availability of new services, more comprehensive care, access to specialised clinicians or psychiatric services, co-location of services, and the destigimitization of services. One external stakeholder illustrated the value of psychiatric services by noting:


*‘Getting a psychiatrist has proved to be the golden egg.’*


Another external stakeholder noted the ability of homelessness services to act as a soft entry point to mental health services through their established relationship with the community, thereby reaching young people who had not previously sought help. Furthermore, they also noted the positive impact of services with peer-workers and services for carers of young people with mental illness.

##### Positive Effects on Consumers and/or Carers

There were two external stakeholders from two Lead Site regions that described the positive effects that the youth enhanced services were having on young consumers including reduced distress, symptomology, and emergency department admissions. Two external stakeholders from two Lead Site regions noted they had received positive feedback from young consumers, carers and practitioners. One external stakeholder said:


*‘Running these groups is very inspiring, enormous sense of relief from the carers. We get no drop-outs. It says a lot—the feedback is extraordinary.’*


Additionally, one external stakeholder reported positive effects of their service for carers of young people with mental health issues. A Lead Site staff member commented that they had seen positive effects on parents following the implementation of a family therapy program.

##### Sector-Wide Impacts

Staff from one Lead Site mentioned strong relationships between services had been developed across the mental health sector. Similarly, two external stakeholders from two Sites commented on improved collaboration between services, which they attributed to the implementation and/or delivery of youth enhanced services:


*‘The key thing for us is collaboration with other providers, such as schools and the justice system.’*



*‘We have been very welcomed by other services that have large waitlists and after a short time, clinical services were keen to work with us.’*


One external stakeholder noted that there has been a shift from activity-based funding towards outcome-based funding, which they regarded as the preferred funding model.

##### Negative Impacts

Staff from two Lead Site regions reported negative impacts of the implementation of youth enhanced services. Specifically, one staff member reported that unsuccessful tenderers for youth enhanced service delivery seemed to have created a more negative view of the PHN commissioning these services. The other staff member noted that stakeholders were concerned about the longevity of the services given there was no guarantee of continued funding by the Australian Government. One staff member also noted that the uncertainty of continued funding created frustration among commissioned service providers and affected the relationship with the PHN.

#### 3.2.3. Future Improvements to Youth Enhanced Service Implementation and Delivery

##### Involving Young People

A major theme that was identified throughout the youth enhanced symposium was the need and opportunity to involve young people with lived experience of mental illness in planning and designing local youth services. A PHN keynote speaker on the youth enhanced symposium commented:


*‘PHNs have an opportunity to connect and partner with young people to co-design local services.’*


##### Improving Service Access

There were six external stakeholders from six Lead Site regions that stated that more or expanded programs are needed to improve access or meet demand in their region. They reported several service gaps that need to be addressed in the future including services for young people with more severe and complex issues, services for those aged under 18 years, programs for parents and carers, specialist programs (e.g., complex trauma), support services (e.g., school engagement), suicide prevention, and after-hours services. One stakeholder noted:


*‘Most services are 09:00 a.m.–5:00 p.m. based, but lots of care is required outside these times. How do we provide effective services out of hours?’*


There were three stakeholders from different regions that highlighted the need for prevention or early intervention services. Lead Site staff added that culturally appropriate services for young people are needed.

##### Addressing Clinical Workforce Shortages

Lead Site staff from two regions noted the need for more ways to attract adequately skilled clinicians for their youth services. Lead Site staff also described a shortage of psychiatrists as a key challenge to the delivery of youth enhanced services. This was echoed by four external stakeholders from three regions. One external stakeholder from a rural city area described getting a local psychiatrist as ‘mission impossible’. A total of seven external stakeholders from five regions also described the need to address workforce recruitment and retention issues more broadly. One external stakeholder noted that some community providers struggle to offer competitive positions:


*‘The service provider did increase the salaries to be a little bit more competitive, but community sector providers don’t come with the benefits of flexibility that either private practice has, or that government employees have. So, it’s difficult for them.’*


##### Improving Data Collection and Usage

There were three external stakeholders from two regions that mentioned challenges that were related to reporting and evaluation requirements including data requirements from both the state and the PHN, having to extract data from different sources, evaluation requirements being disproportionate to funding, and the ‘survey fatigue’ that was experienced by consumers.

One Lead Site staff member suggested that the use of data could be improved to better inform future planning of local youth enhanced services, for example data visualization to communicate with local stakeholders.

##### Establishing Greater Service Integration

Staff from three Lead Sites described the need for greater integration with other youth services. One staff member noted:


*‘Doing everything in isolation we’re not going to achieve anything, we really have to look at how we can work in a much more constructive way together.’*


This was further highlighted during the Lead Site symposium that was attended by PHNs and their service providers. During this symposium the need to provide a continuum of primary mental health services from least to most intensive services to be matched to need (i.e., stepped care approach) was discussed. Improving the ability for young people to step up to, or down from, services as needed was highlighted.

## 4. Discussion

Our findings indicate that effective strategies for the implementation and delivery of youth enhanced services include building upon existing local youth services, establishing effective linkages with other local youth enhanced services and providing complementary non-clinical care. Other effective implementation strategies varied substantially across the different Lead Site regions. This corroborates previous research showing the wide variety of approaches that are taken by several *headspace* centers to establish youth-friendly services [12] and the need for collaboration across mental health services for youth [21]. Notably, both types of stakeholders reported that youth enhanced services adopting a holistic approach which combines clinical and non-clinical services (e.g., vocational or educational support) improved reach to young people with, or at risk of, severe mental illness. This confirms our previous findings in which stakeholders reported that young people respond better when services are integrated and offer non-clinical youth programs complementing clinical services [18]. Furthermore, three Lead Sites reported their service models involved a wrap-around (i.e., multi-disciplinary and collaborative) model that focuses specifically on disengaged young people, incorporating an assertive outreach component. These findings strengthen previous calls for more innovative and flexible mental health service models for young people (i.e., outreach, after-hour services, and various service delivery modes) [7,8]. Ultimately, the current findings indicate that the implementation and delivery of youth mental health services should focus on place-based approaches that are responsive to local population needs, community and sector capacity, and the different local key stakeholders that are involved. The ability to be innovative and flexible in response to region-specific needs and capacities may be a key to success.

Early impacts that were reported by stakeholders indicate that PHN-commissioned youth enhanced services have positive impacts for consumers and carers, such as improved service access and more integrated care. However, future efforts to improve youth enhanced services are still warranted, as shown by the various suggestions for future improvements by both Lead Site staff and external stakeholders. Firstly, stakeholders highlighted the need to consult and co-design youth enhanced services with young people with lived experience of mental illness in the future. The importance of including a wide variety of young people in this process has been illustrated in previous research by the fact that young people have different interpretations and definitions of what is an accessible, acceptable, and appropriate service for them [11]. In particular, it has been pointed out that it is important for youth mental health services to consider whether they are youth-friendly for underrepresented young people and what can be done to engage these groups further [11]. Therefore, youth service providers should aim to consult and co-design youth mental health services with young people, with and without lived experience, and in particular young people from various backgrounds and underrepresented communities in their local regions. There are several precedents on how to involve young people, such as incorporating and responding to end-user feedback, involving youth ambassadors and youth reference groups, or developing a peer support program [22].

Secondly, clinical workforce shortages were especially apparent in consultations with both Lead Sites staff and external stakeholders. This appears to be an ongoing issue, and is also apparent in the earlier stages of youth enhanced service implementation [18]. Telepsychiatry services could potentially offer a (short-term) solution for staff shortages in rural and remote areas [23]. Short-term funding cycles are likely to contribute to the ongoing workforce challenges, as they result in less attractive short-term contracts and ongoing uncertainty for staff members. The negative impacts that were reported by two Lead Site staff members illustrate that short-term funding may have further consequences for Lead Sites and their commissioned services by creating tension in sector relationships and concerns around the longevity of services. This, in turn, may have an impact on local service integration, as this is often dependent on coordination between, and collaboration across, youth mental health services [21]. Finally, ongoing issues around data collection, access, and sharing may be partly overcome by fostering trust between local service providers, the introduction of joint assessments, and/or the introduction of an integrated electronic information system [18,24].

### 4.1. Limitations

These findings should be interpreted considering some limitations. Firstly, the process of implementing youth enhanced services, commissioned by PHN Lead Sites in Australia, is ongoing and long-term with some Lead Sites and external stakeholders still in the earlier stages of service implementation and delivery. Secondly, stakeholders that are participating in the consultations may have provided socially desirable answers but the critical reflections that were offered indicate this was unlikely to be the case. Furthermore, the consultations did not involve young people who received youth enhanced services. It should also be noted that the consultations lacked Aboriginal and Torres Strait Islander representatives (i.e., only one external stakeholder identified as Aboriginal or Torres Strait Islander). As such, the findings do not give any understanding of how PHN commissioned youth enhanced services were received by, or impacted upon, Aboriginal and/ or Torres Strait Islander young people. This represents a significant limitation of the current paper. Finally, it should be noted that in order to evaluate the effectiveness on young people’s mental health and wellbeing of these youth enhanced services, future research will need to study the treatment outcomes of these services.

### 4.2. Conclusions

Overall, these findings highlight the necessity for collaborative and localized responses as well as service models that combine clinical and non-clinical care to address the needs of young people with, or at risk of, severe mental illness. Early impacts that were reported by stakeholders indicated that such youth enhanced services models have positive impacts for consumers, carers, and the wider service sector. Future efforts to improve youth enhanced services are warranted and should focus on involving young people in service planning and design, expanding programs to address local needs, developing and recruiting appropriate (clinical) workforce, improving data collection and usage, and facilitating service integration.

## Data Availability

The data that support the findings of this study are not publicly available and restriction apply. Data are available from the authors upon reasonable request and with permission of the Australian Government Department of Health.

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
