# Peer review of "Implementing and Delivering Youth Mental Health Services: Approaches Taken by the Australian Primary Health Network ‘Lead Sites’"

_ijerph, 2022, doi:10.3390/ijerph191710494_

Round 1

Reviewer 1 Report

  1. Implementing and delivering youth mental health services: Approaches taken by the Australian is a preliminary model with lead plan model.

  2. The paper needs revision. Modify Introduction with more focussed on the project.
  3. Modify the methods with more details
  4. Conclusion needs to be more focussed.
  5. The paper is well written.
  1.  

  2.  

Author Response

  • The paper needs revision. Modify Introduction with more focussed on the project.
  • Modify the methods with more details
  • Conclusion needs to be more focussed.
  • The paper is well written.

Response:

The authors thank the reviewer for their time and effort. We have revised and restructured the introduction, restructured and added more details to the method section, and restructured and provided more focus to the discussion and conclusion.

Reviewer 2 Report

The manuscript is poorly conceptualized and drafted. There are major flaws in the manuscript, a few to mention are as below.

1. The objective of the study is not clearly articulated.

2.  Study population along with exclusion and inclusion criteria is missing.

3.  Most of the statements are generalized in nature, like, "We used several qualitative data sources collected from September 2018 to March 2019", the author should have clearly mentioned the data sources, how the data was collected, and how the data was processed and analyzed. 

4. The authors failed to mention how the themes were extracted. 

The manuscript also lacks a logical flow. Some parts of the methods are mentioned in the introduction part, while some parts are mentioned in the results part.

These are a few major points. Similar flaws are present in the manuscript. 

Author Response

The manuscript is poorly conceptualized and drafted. There are major flaws in the manuscript, a few to mention are as below.

  • The objective of the study is not clearly articulated.

Response:

The authors thank the reviewer for their time and effort. We have clarified the aim at the end of the introduction as follows:

‘To inform a better understanding of effective implementation of mental health services for young people, this paper reports on effective approaches for, and early impacts of, implementing and delivering of youth enhanced services commissioned by 10 PHN Lead Sites (reform leaders) in Australia.’

  • Study population along with exclusion and inclusion criteria is missing.

Response:

The study population was described for PHN staff and external stakeholders in separate section, we have restructured the paragraph to clarify the inclusion criteria:

Semi-structured consultations were held with staff from Lead Sites and external stakeholders (e.g., individual providers or service provider representatives) in Lead Site regions involved in the planning, implementation and/or delivery of PHN commissioned services. Lead Site representatives received the consultation questions in advance of the consultations and were asked to invite participation of those staff best able to answer the questions. Lead Site representatives were asked to review the list of 10 to 20 external regional stakeholders who participated in an earlier round of consultations(18), to revise their contact details and provide details for additional stakeholders from their region.

To further clarify exclusion criteria, we added:

‘There were no exclusion criteria for participation in the consultations.’

  • Most of the statements are generalized in nature, like, "We used several qualitative data sources collected from September 2018 to March 2019", the author should have clearly mentioned the data sources, how the data was collected, and how the data was processed and analyzed.

Response:

We have adjusted the statement on data sources to clarify that these are described in more detail in the subsequent section:

‘Several qualitative data sources were used, which were collected from September 2018 to March 2019. These included stakeholder consultations with Lead Site staff and external stakeholders in Lead Site regions, and observational data from one Lead Site meeting and a national Lead Site symposium. These are discussed in more detail below.’

We have also added more details on how data was processed and analysed:

‘Data from the consultations with Lead Site staff and external stakeholders and the observational data, were anaysed using NVivo 12. Each transcript was coded with an initial template based on the pre-determined questions focusing on youth enhanced services (Appendix 1). Two researchers deductively identified data related to the implementation and delivery of youth enhanced services using a thematic analysis approach (19). Each researcher coded one transcript independently, iteratively creating a coding template. Once the final coding template was constructed, the transcripts were re-examined and the researchers inductively identified narrower sub-themes. A framework analysis approach was used to order the themes in an overarching framework to make sense of the emerging themes (20). The two researchers compared and revised the coding template until consensus was reached. The final was applied across all relevant stakeholder data.’

Th described thematic approach has been previously adopted in the other previous articles such as:

Oostermeijer, S., Bassilios, B., Nicholas, A., Williamson, M., Machlin, A., Harris, M., ... & Pirkis, J. (2021). Implementing child and youth mental health services: early lessons from the Australian Primary Health Network Lead Site Project. International Journal of Mental Health Systems, 15(1), 1-13.

King, K., Hall, T., Oostermeijer, S., & Currier, D. (2022). Community participation in Australia’s National Suicide Prevention Trial. Australian journal of primary health.

  • The authors failed to mention how the themes were extracted.

We added the additional details, also mentioned above:

‘..and the researchers inductively identified narrower sub-themes. A framework analysis approach was used to order the themes in an overarching framework to make sense of the emerging themes (20).’

  • The manuscript also lacks a logical flow. Some parts of the methods are mentioned in the introduction part, while some parts are mentioned in the results part.

We have made adjustments throughout the manuscript to improve the flow and the following section has been moved from introduction to method section:

‘Preliminary findings discussing main challenges of, and facilitating factors to, implementing youth enhanced services during the first year and a half (from 1 July 2016 to 31 December 2017) of PHN activities have been described elsewhere (18). The current study involves data collection approximately one year later and describes the effective approaches for, and early impacts of, the implementation and delivery of youth enhanced services in Australia

Reviewer 3 Report

Thank for your allowing me the opportunity to review this manuscript. Qualitative research of this kind is essential to understanding the needs of local mental health services as well as stakeholders requiring those services. I believe this publication adds new information to the debate regarding service gaps and the ‘missing middle’ that is so abundant in the youth mental health service industry.

To editor:

Line 141 – Were the researchers trained in themed notetaking?

Line 145 – What are the expertise of the researchers involved in the thematic analysis approach?

Line 154 – The Lack of Aboriginal & Torres Strait Islander representation causes a significant gap in the understanding of how these services do or do not appeal to youth of Aboriginal or Torres Strait Islander descent. This should be added to the report’s limitations.

Line 356/360 – is there a procedure for onboarding youth with lived experience into the service planning and design of programs? Although strongly supported, unless the authors have some idea as to how to access, approach, and on-board these young people, this suggestion feels somewhat hollow. Were there any responses from lead site staff or stakeholders regarding this process?

Author Response

  • Line 141 – Were the researchers trained in themed notetaking?

Response:

The authors thank the reviewer for their time and effort. They were note trained but had previous experience, we added the following:

‘with experience in themed notetaking’

  • Line 145 – What are the expertise of the researchers involved in the thematic analysis approach?

Response:

  • All authors have substantial experience in qualitative research using a thematic analysis approach. This is demonstrated by several other publications for example:

Oostermeijer, S., Dwyer, M., & Tongun, P. (2022). Relational security: The impact of facility design on youth custodial staffs' practices and approaches. Criminal Behaviour and Mental Health.

Oostermeijer, S., Bassilios, B., Nicholas, A., Williamson, M., Machlin, A., Harris, M., ... & Pirkis, J. (2021). Implementing child and youth mental health services: early lessons from the Australian Primary Health Network Lead Site Project. International Journal of Mental Health Systems, 15(1), 1-13.

Bassilios, B., Nicholas, A., Ftanou, M., Fletcher, J., Reifels, L., King, K., ... & Pirkis, J. (2017). Implementing a primary mental health service for children: Administrator and provider perspectives. Journal of Child and Family Studies, 26(2), 497-510.

  • Line 154 – The Lack of Aboriginal & Torres Strait Islander representation causes a significant gap in the understanding of how these services do or do not appeal to youth of Aboriginal or Torres Strait Islander descent. This should be added to the report’s limitations.

Response:

We added the following to the discussion:

‘It should also be noted that the consultations lacked Aboriginal and Torres Strait Islander representatives (i.e. only one external stakeholder identified as Aboriginal or Torres Strait Islander). As such, the findings do not give any understanding of how PHN commissioned youth enhanced services were received by, or impacted upon, Aboriginal and/ or Torres Strait Islander young people. This represents a significant limitation of the current paper.’

  • Line 356/360 – is there a procedure for onboarding youth with lived experience into the service planning and design of programs? Although strongly supported, unless the authors have some idea as to how to access, approach, and on-board these young people, this suggestion feels somewhat hollow. Were there any responses from lead site staff or stakeholders regarding this process?

Response:

There was no data identified that related to how to access, approach, or involve young people with lived experience in the service planning or design of programs.

We added the following to the discussion:

‘There are several precedents on how to involve young people, such as incorporating and responding to end-user feedback, involving youth ambassadors and youth reference groups, or developing a peer support program (22).’
